# ICU-acquired infections and mortality in community-acquired pneumonia–induced sepsis: Insights from a transcriptomic analysis

Sultan Almuntashiri[1], Yazed S. Alsowaida[1], Duo Zhang[2,3], Ali Alghubayshi[1], Muteb Alanazi[1], Tareq Nafea Alharby[1], Mukhtar Ansari[1]*

1 Department of Clinical Pharmacy, College of Pharmacy, University of Ha'il, Ha'il, Saudi Arabia,
2 Department of Clinical and Administrative Pharmacy, College of Pharmacy, University of Georgia, Augusta, Georgia, United States of America, 3 Clinical and Experimental Therapeutics, College of Pharmacy, University of Georgia and Charlie Norwood VA Medical Center, Augusta, Georgia, United States of America

* mukhtaransari@hotmail.com

## Abstract

### Objectives

To evaluate the association between ICU-acquired infections and 28-day mortality in pneumonia-induced sepsis and to explore associated immune-related gene expression patterns.

### Methods

A secondary analysis was performed using the publicly available GSE65682 dataset, including adult ICU patients with sepsis secondary to community-acquired pneumonia (CAP) or hospital-acquired pneumonia (HAP). Patients were stratified based on the development of ICU-acquired infections. 28-day mortality and whole-blood leukocyte gene expression at ICU admission were compared between groups.

### Results

Among 144 patients, 20 developed ICU-acquired infections. In the CAP subgroup, ICU-acquired infections were associated with numerically higher 28-day mortality compared to those without infection (45.5% vs. 18.2%, p = 0.05), although this finding should be interpreted with caution given the retrospective study design and limited sample size. In HAP patients, a similar pattern was not observed (22.2% vs. 19.6%). Transcriptomic analysis showed significant downregulation of the interleukin-7 receptor (IL7R) in CAP patients who developed ICU-acquired infections, with PRKACB and CD3D also demonstrating a downward trend.

**Data availability statement:** The original transcriptomic dataset is publicly available in the Gene Expression Omnibus under accession number GSE65682. (https://www.ncbi.nlm.nih.gov/geo/query/acc.cgi?acc=GSE65682).

**Funding:** This research is funded by Scientific Research Deanship at University of Ha'il- Saudi Arabia through project number BA-24 029. The funders had no role in study design, data collection and analysis, decision to publish, or preparation of the manuscript.

**Competing interests:** The authors have declared that no competing interests exist.

## Conclusion

These findings suggest that early immune dysregulation may be associated with an increased susceptibility to secondary infections and potentially worse outcomes among CAP patients. IL7R may represent a candidate signal of immune dysregulation and warrants further investigation and validation in future studies.

## Introduction

Sepsis remains a major contributor to morbidity and mortality among critically ill patients, particularly in the intensive care unit (ICU). Despite gradual global improvements in outcomes, sepsis continues to affect approximately 49 million people annually and is responsible for nearly 11 million deaths. Mortality may reach up to 60% in patients who progress to septic shock, a severe subset marked by profound circulatory and metabolic disturbances [1,2]. Sepsis represents a complex dysregulation of the immune system, characterized by a simultaneous activation of pro-inflammatory and anti-inflammatory pathways. This immunological imbalance triggers a widespread release of cytokines, host-derived mediators, and pathogen-associated signals ultimately driving systemic inflammation and multi-organ dysfunction [3,4].

Community-acquired pneumonia (CAP) is a leading infectious cause of sepsis, particularly among hospitalized and critically ill patients [5]. In the ICU setting, CAP-induced sepsis presents unique challenges due to its rapid progression, heterogeneity in host response, and high risk for complications, including secondary infections and organ failure [6,7]. Despite advances in antimicrobial therapy and supportive care, CAP remains associated with substantial morbidity and mortality when it progresses to sepsis. The underlying immune dysregulation in CAP-related sepsis may contribute to increased susceptibility to ICU-acquired infections, yet the molecular mechanisms contributing to this susceptibility remain poorly understood.

ICU-acquired infections are common complications among critically ill patients, particularly those with sepsis [8]. These infections often arise after several days of ICU stay and are associated with prolonged mechanical ventilation, immunosuppression, and exposure to invasive devices [9]. Patients with CAP-induced sepsis may be especially vulnerable due to the initial dysregulated immune response and impaired barrier defenses. ICU-acquired infections not only increase the risk of organ dysfunction and mortality but also complicate treatment strategies [10]. Understanding the biological factors associated with ICU-acquired infections is crucial for identifying high-risk individuals and improving outcomes.

Gene expression profiling offers a powerful approach to understanding disease susceptibility in critically ill patients. By analyzing transcriptional activity in whole blood, researchers can uncover immune-related pathways linked to clinical outcomes [11]. While prior studies have examined transcriptomic patterns in sepsis, few have focused on gene-level differences associated with ICU-acquired infections—particularly in patients with CAP-induced sepsis. In this study, we aimed to evaluate the association between ICU-acquired infections and 28-day mortality in patients with

pneumonia-induced sepsis and to explore associated immune-related gene expression profiles, with the goal of identifying potential molecular signals of infection risk and host immune vulnerability.

## Materials and methods

### Study design and data source

This was a retrospective secondary analysis of the publicly available gene expression dataset GSE65682, obtained from the Gene Expression Omnibus (GEO) [12]. The dataset includes whole-blood RNA samples collected from adult ICU patients admitted with sepsis or non-infectious critical illness. Blood samples were collected using PAXgene tubes at ICU admission and during the ICU stay. Transcriptomic profiling was performed using genome-wide microarrays.

From the original dataset, we included only patients with sepsis due to pneumonia, specifically those diagnosed with CAP or hospital-acquired pneumonia (HAP). We then identified patients who developed an ICU-acquired infection during their ICU stay. The final analysis included both survival comparisons and gene expression analysis between ICU-acquired infection and non–ICU-acquired infection patients. These comparisons were performed in the overall cohort, as well as within the CAP and HAP subgroups.

ICU-acquired infection status was obtained from the clinical annotation variable provided in the GSE65682 dataset. In the original MARS cohort, ICU-acquired infections were defined as infections occurring more than 48 hours after ICU admission.

### Gene expression processing and statistical analysis

Raw microarray data were processed as described in the original GSE65682 dataset. Briefly, array data were background corrected using Robust Multi-array Average (RMA), quantile normalized, and summarized using median polish. Probe intensities were filtered using a variance cutoff of 0.5 to retain 24,646 expressed probes. Potential batch effects were evaluated using Surrogate Variable Analysis and corrected using the ComBat empirical Bayes method.

Candidate genes in the present analysis were pre-specified based on previously reported roles in immune regulation and sepsis-related host response in the literature. Because the present analysis focused on a small number of biologically relevant candidate genes rather than genome-wide discovery, the transcriptomic analysis was considered exploratory. Therefore, formal correction for multiple comparisons was not applied, as the analysis was hypothesis-driven and limited to a small number of pre-specified candidate genes.

Descriptive statistics were used to summarize patient demographics and outcomes. Differences in gene expression between groups were analyzed using the Mann–Whitney U test. Kaplan–Meier survival curves were generated to compare 28-day survival between groups, and the log-rank test was used for significance testing. Effect sizes for mortality comparisons were expressed as odds ratios with 95% confidence intervals. Multivariable logistic regression analysis was performed, adjusting for available baseline variables including age, gender, and diabetes mellitus. Given the absence of severity indices in the dataset and other important confounders, adjustment for illness severity, ICU interventions and comorbidities was not possible. A $p$-value of less than or equal to 0.05 was considered statistically significant. Statistical analysis was performed using SPSS version 30, and all figures, including gene expression data and Kaplan–Meier survival curves, were created using GraphPad Prism version 10.

### Ethics statement

This study analyzed publicly available, de-identified transcriptomic data (GSE65682) obtained from the Gene Expression Omnibus (GEO). Ethical approval for the original MARS consortium study was obtained by the original investigators, and informed consent was obtained from patients or their representatives. No additional ethical approval or consent was required for this secondary analysis of de-identified data.

## Results

### Baseline demographic and clinical characteristics of included patients

A total of 144 ICU patients with pneumonia induced sepsis were included, of whom 79 (54.9%) had CAP and 65 (45.1%) had HAP. The mean age was similar between groups (61.1 ± 16.3 years in CAP vs. 59.2 ± 15.1 years in HAP; $p = 0.46$). There were no statistically significant differences in gender distribution (male: 55.7% vs. 63.1%, $p = 0.37$), prevalence of diabetes mellitus (24.1% vs. 13.8%, $p = 0.12$), or 28-day mortality rates (22.1% vs. 20.0%, $p = 0.76$) between the CAP and HAP-induced sepsis groups (Table 1).

### ICU-acquired infections

Among the 144 ICU patients, a total of 20 patients (13.9%) developed ICU-acquired infections. There were no statistically significant differences in age (59.7 ± 18.3 vs. 60.3 ± 15.4 years, $p = 0.87$), gender distribution (male: 75% vs. 56.5%, $p = 0.12$), or diabetes prevalence (20.0% vs. 19.4%, $p = 0.94$) between patients with and without ICU-acquired infections.

Later, we stratified the patients by pneumonia type to better assess group-specific outcomes. In HAP, a total of 9 patients (13.8%) developed ICU-acquired infection. Although the proportion of males was higher in the infected group (88.9% vs. 58.9%), the difference was not statistically significant ($p = 0.08$). Age and diabetes were comparable between groups. In CAP, a total of 11 patients (13.9%) developed ICU-acquired infections. No significant differences were observed in age ($p = 0.63$), gender ($p = 0.56$), or diabetes ($p = 0.78$) between infected and non-infected groups (Table 2).

### 28-day mortality outcomes

The 28-day mortality rate did not differ significantly between patients with and without ICU-acquired infections in the overall cohort (35.0% vs. 18.8%; log-rank p = 0.149) (Fig 1A). Later, we stratified the patients by pneumonia type. Among patients with HAP, mortality was similar between those with and without ICU-acquired infections (22.2% vs. 19.6%; log-rank p = 0.94; Fig 1B). In contrast, CAP patients who developed ICU-acquired infections had numerically higher mortality compared to those without (45.5% vs. 18.2%; log-rank p = 0.05; Fig 1C).

### Logistic regression analyses for 28-day mortality

To further evaluate the association between ICU-acquired infections and 28-day mortality, we performed unadjusted and adjusted logistic regression analyses (Table 3). In the overall cohort, ICU-acquired infection was not significantly associated with mortality in both unadjusted (OR 2.31, 95% CI 0.70–7.14, p = 0.136) and adjusted analyses (OR 2.57, 95% CI 0.84–7.50, p = 0.087). Similar findings were observed in the HAP subgroup, where ICU-acquired infection was not associated with mortality in either model. In contrast, among CAP patients, ICU-acquired infection was associated with higher

**Table 1. Baseline demographic and clinical characteristics of patients with CAP and HAP induced sepsis.**

| Variable | All (n = 144) | CAP (n = 79) | HAP (n = 65) | p-value |
|---|---|---|---|---|
| **Demographics** | | | | |
| **Age (mean ± SD)** | 60.22 (15.77) | 61.09 (16.31) | 59.15 (15.14) | 0.46 |
| **Gender male, n (%)** | 85 (59) | 44 (55.7) | 41 (63.1) | 0.37 |
| **Clinical Characteristics** | | | | |
| **Diabetes mellitus (%)** | 28 (19.4) | 19 (24.1) | 9 (13.8) | 0.12 |
| **Mortality at 28 days (%)** | 29 (21.2) | 17 (22.1) | 12 (20) | 0.76 |

Continuous variables are presented as mean ± standard deviation, and categorical variables are shown as counts (percentages). P-values were calculated using t-tests or chi-square tests, as appropriate.

**Table 2. Comparison of demographic and clinical characteristics between patients with and without ICU-acquired infections.**

| Variable | All patients (n = 144) | ICU-Acquired Infection (n = 20) | No ICU-Acquired Infection (n = 124) | p-value |
|---|---|---|---|---|
| Age (mean ± SD or median) | 60.22 (15.77) | 59.7 (18.27) | 60.3 (15.41) | 0.87 |
| Gender (male, %) | 85 (59) | 15 (75) | 70 (56.5) | 0.12 |
| Diabetes mellitus (%) | 28 (19.4) | 4 (20.0) | 24 (19.4) | 0.94 |
| Variable | HAP patients (n = 65) | ICU-Acquired Infection (n = 9) | No ICU-Acquired Infection (n = 56) | p-value |
| Age (mean ± SD or median) | 59.15 (15.14) | 60.67 (15) | 58.91 (15.28) | 0.75 |
| Gender (male, %) | 41 (63.1) | 8 (88.9) | 33 (58.9) | 0.08 |
| Diabetes mellitus (%) | 9 (13.8) | 1 (11.1) | 8 (14.3) | 0.79 |
| Variable | CAP patients (n = 79) | ICU-Acquired Infection (n = 11) | No ICU-Acquired Infection (n = 68) | p-value |
| Age (mean ± SD or median) | 61.09 (16.31) | 58.91 (21.28) | 61.44 (15.53) | 0.63 |
| Gender (male, %) | 44 (55.7) | 7 (63.6) | 37 (54.4) | 0.56 |
| Diabetes mellitus (%) | 19 (24.1) | 3 (27.3) | 16 (23.5) | 0.78 |

Data are expressed as mean ± standard deviation for continuous variables and number (percentage) for categorical variables. P-values were calculated using appropriate statistical tests.

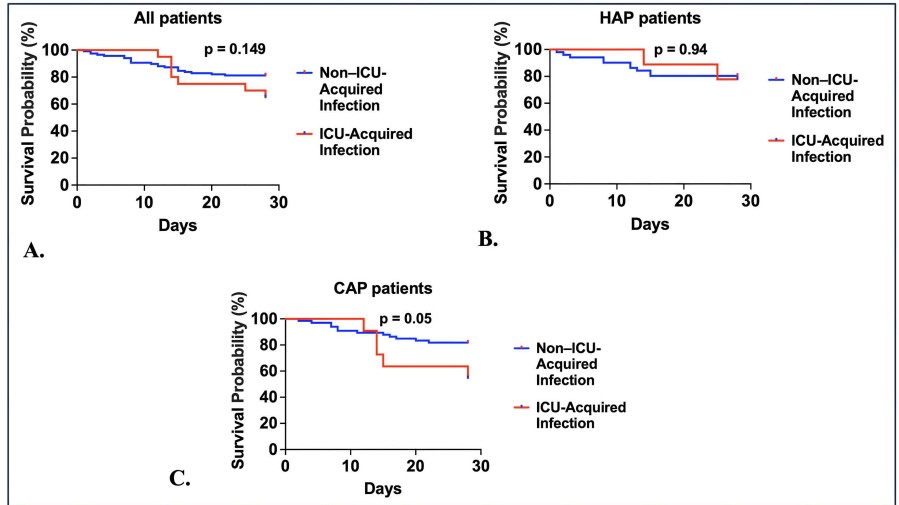

**Fig 1. 28-day mortality among patients with and without ICU-acquired infections, stratified by pneumonia type.** Kaplan–Meier survival analysis for ICU patients with sepsis, censored at 28 days, comparing those who developed ICU-acquired infections versus those who did not. Analyses were conducted for **(A)** all patients, **(B)** patients with hospital-acquired pneumonia (HAP), and **(C)** patients with community-acquired pneumonia (CAP). The log-rank test was used to compare survival distributions.

mortality in the adjusted analysis (OR 4.97, 95% CI 1.14–23.40, p = 0.033), while the unadjusted analysis showed a similar trend that did not reach statistical significance (OR 3.67, 95% CI 0.75–17.32, p = 0.058). Other variables, including age, gender, and diabetes mellitus, were not significantly associated with mortality across all models.

## Differential gene expression analysis

To explore whether immune-related gene expression patterns differ between patients who develop ICU-acquired infections and those who do not, we assessed the expression levels of five pre-specified immune-related genes. This analysis was conducted in the overall cohort and further stratified by pneumonia type. (Fig 2A–O). Protein kinase, cAMP-dependent,

**Table 3. Logistic regression analyses for 28-day mortality in all patients, HAP, and CAP subgroups.**

| Variable | OR | 95% CI | p-value |
|---|---|---|---|
| **All patients (Unadjusted)** | | | |
| ICU-acquired infection | 2.31 | 0.70–7.14 | 0.136 |
| **All patients (Adjusted)** | | | |
| ICU-acquired infection | 2.57 | 0.84–7.50 | 0.087 |
| Age | 1.02 | 1.00–1.06 | 0.123 |
| Gender (Male) | 1.22 | 0.51–2.92 | 0.654 |
| Diabetes mellitus | 0.32 | 0.07–1.08 | 0.095 |
| **HAP (Unadjusted)** | | | |
| ICU-acquired infection | 1.17 | 0.10–7.56 | 0.99 |
| **HAP (Adjusted)** | | | |
| ICU-acquired infection | 1.18 | 0.15–6.54 | 0.855 |
| Age | 1.01 | 0.97–1.06 | 0.536 |
| Gender (Male) | 1.18 | 0.29–4.65 | 0.811 |
| Diabetes mellitus | 0.41 | 0.02–2.82 | 0.437 |
| **CAP (Unadjusted)** | | | |
| ICU-acquired infection | 3.67 | 0.75–17.32 | 0.058 |
| **CAP (Adjusted)** | | | |
| ICU-acquired infection | 4.97 | 1.14–23.40 | 0.033 |
| Age | 1.03 | 1.00–1.08 | 0.110 |
| Gender (Male) | 1.02 | 0.30–3.33 | 0.971 |
| Diabetes mellitus | 0.24 | 0.03–1.10 | 0.102 |

Patients with hospital-acquired pneumonia (HAP); Patients with community-acquired pneumonia (CAP).
Adjusted models include age, gender, and diabetes mellitus. OR = odds ratio; CI = confidence interval.

catalytic, beta (PRKACB), a key regulator of intracellular signaling pathways involved in immune cell activation and cytokine production [13], was expressed at lower levels in patients with ICU-acquired infections across all groups, with the most pronounced reduction observed in CAP patients (Fig 2A–C). CASP8 and FADD-like apoptosis regulator (CFLAR), an anti-apoptotic molecule involved in regulating cell death and immune signaling [14], showed consistently decreased expression in the ICU-acquired infection group, particularly among CAP patients (Fig 2D–F). CD3D, a component of the T-cell receptor complex essential for T-cell activation and immune signaling [15], showed lower expression in ICU-acquired infection patients, especially within the CAP subgroup, although this difference did not reach statistical significance ($p = 0.054$) (Fig 2G–I). Tumor necrosis factor, alpha-induced protein 6 (TNFAIP6), a gene involved in resolving inflammation and maintaining extracellular matrix stability [16], exhibited elevated expression in patients with ICU-acquired infections, particularly among those with CAP (Fig 2J–L). Interleukin 7 receptor (IL7R), a critical regulator of lymphocyte development and survival [17], showed significantly reduced expression in patients with ICU-acquired infections in the overall cohort ($p = 0.04$), with a more pronounced significant difference observed among CAP patients ($p = 0.01$) (Fig 2M–O). Fold change analysis showed modest differences in gene expression between groups. In CAP patients, IL7R, PRKACB, and CD3D demonstrated more pronounced downregulation (fold changes 0.84, 0.89, and 0.86, respectively), while changes in the overall cohort and HAP subgroup were smaller and less consistent (Table 4).

## Discussion

In this study, we investigated the potential association between ICU-acquired infections and 28-day mortality in patients with pneumonia-induced sepsis and explored associated immune-related gene expression patterns. We observed that

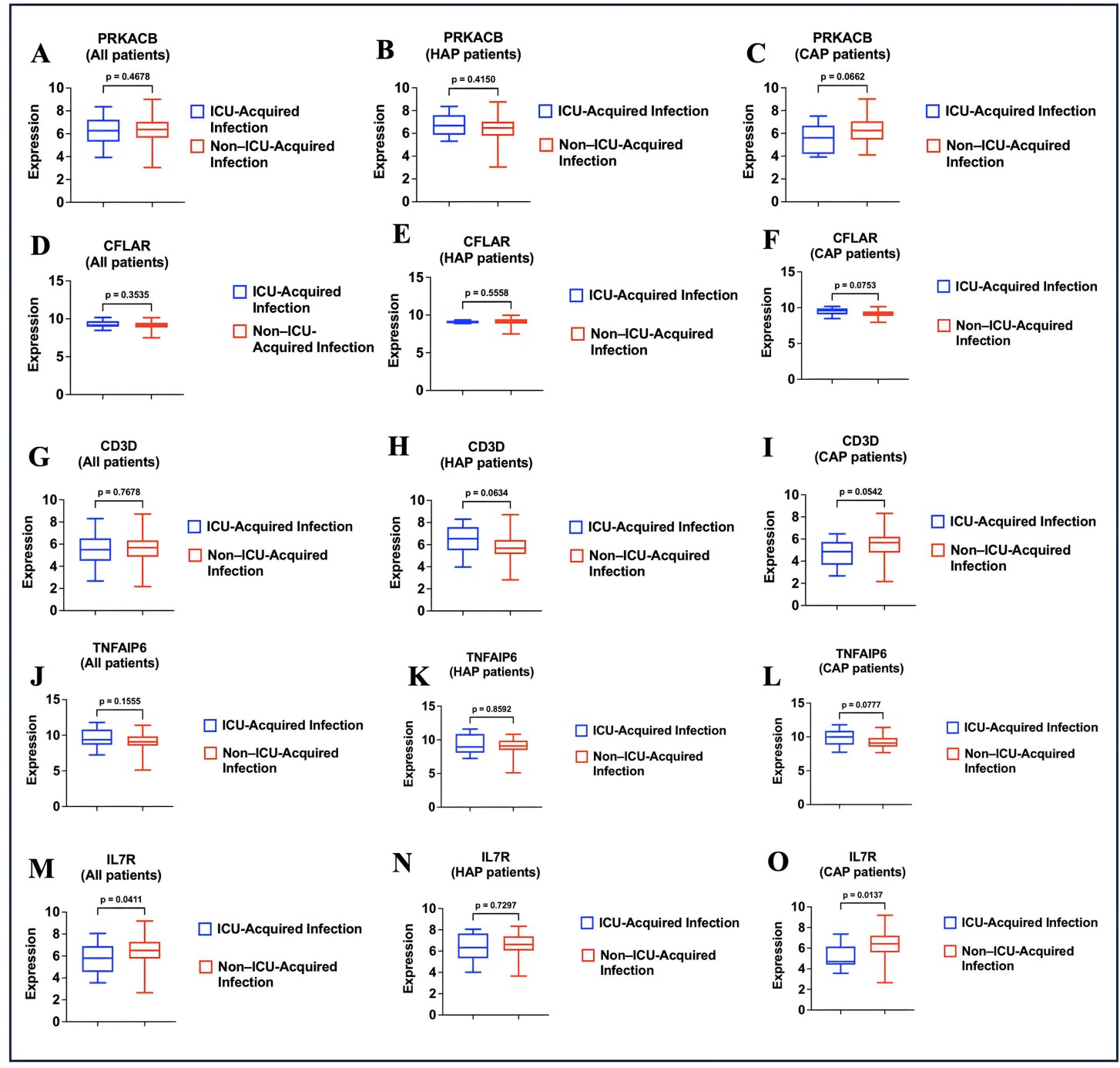

**Fig 2. Differential expression of immune-related genes in ICU-acquired vs. non–ICU-acquired infections, stratified by pneumonia type. PRKACB:** Protein kinase, cAMP-dependent, catalytic, beta; **CFLAR:** CASP8 and FADD-like apoptosis regulator; **CD3D:** CD3d molecule, delta (CD3-TCR complex); **TNFAIP6:** Tumor necrosis factor, alpha-induced protein 6; **IL7R:** Interleukin 7 receptor. **CAP:** Community acquired pneumonia; **HAP:** Hospital acquired pneumonia. **All patients** (n = 144); ICU-Acquired Infection (n = 20); No ICU-Acquired Infection (n = 124); **HAP patients** (n = 65); ICU-Acquired Infection (n = 9); No ICU-Acquired Infection (n = 56); **CAP patients** (n = 79); ICU-Acquired Infection (n = 11); No ICU-Acquired Infection (n = 68).

**Table 4. Fold Change of Immune-Related Genes.**

| Group | Gene | Fold Change |
|---|---|---|
| **All patients** | IL7R | 0.90 |
| | PRKACB | 0.96 |
| | CD3D | 0.99 |
| | CFLAR | 1.02 |
| | TNFAIP6 | 1.05 |
| **HAP patients** | IL7R | 0.97 |
| | PRKACB | 1.05 |
| | CD3D | 1.15 |
| | CFLAR | 1.00 |
| | TNFAIP6 | 1.03 |
| **CAP patients** | IL7R | 0.84 |
| | PRKACB | 0.89 |
| | CD3D | 0.86 |
| | CFLAR | 1.03 |
| | TNFAIP6 | 1.07 |

Fold change represents the ratio of mean gene expression in patients with ICU-acquired infections to those without infection. Values <1 indicate downregulation, while values >1 indicate upregulation. **IL7R:** Interleukin 7 receptor**; PRKACB:** Protein kinase, cAMP-dependent, catalytic, beta; **CD3D:** CD3d molecule, delta (CD3-TCR complex); **CFLAR:** CASP8 and FADD-like apoptosis regulator; **TNFAIP6:** Tumor necrosis factor, alpha-induced protein 6. **CAP:** Community acquired pneumonia; **HAP:** Hospital acquired pneumonia.

CAP patients who developed ICU-acquired infections had higher odds of 28-day mortality in adjusted analysis; however this finding should be interpreted cautiously given the observational study design, small sample size, wide confidence intervals and the lack of adjustment for important confounders, including illness severity, ICU interventions and comorbidities. Transcriptomic analysis revealed distinct immune alterations among CAP patients, including reduced expression of PRKACB, CD3D, and IL7R, genes involved in intracellular signaling, T-cell activation, and lymphocyte survival. Given that gene expression was measured at ICU admission, reveres casualty cannot be excluded limiting causal interpretation. Moreover, the transcriptomic findings should be considered hypothesis-generating given the exploratory nature of the analysis, the use of a small number of pre-specified genes, and the absence of multiple testing correction.

Our findings show that patients with CAP–induced sepsis who developed ICU-acquired infections later had numerically higher 28-day mortality. In contrast, this association was not observed in patients with HAP–induced sepsis. These results suggest that the clinical consequences of ICU-acquired infections may vary depending on the source of the initial infection. CAP patients, who often present from the community with no prior antibiotic exposure or critical illness, may experience more pronounced immune disruption when faced with a second infection during their ICU stay. This difference in mortality patterns between CAP and HAP may suggest the need for further investigation into subgroup-specific risk. This differential mortality pattern between CAP- and HAP-induced sepsis appears to be relatively underexplored in the current literature.

Among the immune-related genes assessed, IL7R was the only gene that showed statistically significant downregulation in patients with ICU-acquired infections, particularly in those with CAP-induced sepsis. Given IL7R's pivotal role in maintaining lymphocyte homeostasis, its reduced expression may be associated with persistent lymphopenia, impaired immune recovery, and heightened susceptibility to secondary infections in the ICU setting [18]. Prior studies have shown

that lower IL7R expression is associated with increased mortality in septic shock and has been proposed as a potential signal for identifying patients who may benefit from immunoadjuvant therapies, such as recombinant IL-7 [19]. In randomized trials, recombinant IL-7 has been shown to be beneficial in restoring T-cell counts in septic patients with lymphopenia [20,21]. Importantly, a study in hospitalized CAP patients without sepsis reported that low IL7R expression was associated with ICU admission and 30-day mortality [22]. The observed downregulation of IL7R in our CAP-induced sepsis subgroup is consistent with the hypothesis that immune suppression may underlie their heightened mortality risk following ICU-acquired infections and suggests that IL7R may represent a candidate immune signal warranting further investigation and validation in future studies.

In addition to IL7R, two other genes—PRKACB and CD3D—showed consistent downregulation in patients with ICU-acquired infections, particularly among those with CAP-induced sepsis. Although these changes did not reach statistical significance, the downward trend in expression may still hold biological relevance. In a bioinformatics analysis of pediatric and adult sepsis datasets [23], PRKACB was identified as one of the top diagnostic genes, with a high area under the curve (AUC) for differentiating sepsis and septic shock. Its expression exhibited a progressive dysregulation pattern correlating with increasing disease severity. Supporting this, another study demonstrated that PRKACB was significantly downregulated in plasma extracellular vesicles of sepsis patients [24]. These findings are consistent with our current observation of reduced PRKACB expression in CAP-induced sepsis patients with ICU-acquired infections, which may further suggest a potential link between this gene and immune dysregulation associated with secondary infection. Regarding CD3D, reduced expression has been associated with impaired T-cell receptor signaling and worse clinical outcomes in sepsis, suggesting its potential role as a signal of immune suppression in critically ill patients [25]. Additionally, CD3D has been identified as a diagnostic candidate gene in COVID-19 patients with sepsis, reinforcing its relevance in immune dysregulation [26]. Consistent with these findings, our analysis also showed lower CD3D expression in patients with ICU-acquired infections, particularly in the CAP subgroup, which may further suggest its potential role as a signal of immunosuppression in this vulnerable population.

A major strength of this study lies in its integrated approach, combining clinical outcomes with transcriptomic profiling to enhance our understanding of ICU-acquired infections in pneumonia-induced sepsis. By specifically analyzing patients with CAP and HAP, we were able to observe differences in mortality patterns and immune gene expression between these subgroups. Stratifying by pneumonia type and secondary infection status allowed for an exploratory characterization of patients and their associated molecular signatures. These findings may contribute to better understanding of immune dysregulation in the population. IL7R may represent a candidate signal requiring further investigation and validation in future studies.

## Limitations

This study has several limitations that should be acknowledged. First, it was a retrospective observational analysis of an existing dataset. Second, the publicly available dataset did not include established severity indices such as SOFA or APACHE II. Therefore, multivariable logistic regression was performed using only the available variables (age, gender and diabetes mellitus). Adjustment for important confounders was limited, which may have resulted in residual confounding. In particular, the potential influence of unmeasured confounding factors, including antibiotic timing, disease severity, comorbidities and ICU-related interventions, cannot be excluded. Third, gene expression was measured at the mRNA level and may not fully reflect protein-level activity or downstream immune function. Fourth, the sample size—particularly in the subgroup of CAP patients with ICU-acquired infections—was relatively small, which may limit statistical power and the stability of these estimates, and the findings should therefore be interpreted cautiously. Fifth, because gene expression was measured at ICU admission, the temporal relationship between gene expression and subsequent ICU-acquired infections remains uncertain, and reverse causality cannot be excluded, limiting causal interpretation of the findings. Additionally, the absence of formal correction for multiple testing may increase the risk of false-positive findings. Finally, external validation in an independent cohort was not performed, and future studies are needed to validate these findings in larger cohorts.

## Conclusions

Our study suggests a clinically and biologically distinct risk profile among CAP-induced sepsis patients who develop ICU-acquired infections. The combination of numerically higher mortality and altered immune gene expression may suggest a vulnerable subgroup that may warrant further investigation. Future research should validate these findings in larger, prospective cohorts and explore the potential of immune-related signals in better understanding risk stratification and immune dysregulation in ICU patients.

## Supporting information

**S1 Checklist. STROBE checklist.**
(DOC)

## Acknowledgments

The authors acknowledge the Scientific Research Deanship at the University of Ha'il – Saudi Arabia for supporting this work through project number BA-24–029.

## Author contributions

**Conceptualization:** Sultan Almuntashiri.

**Data curation:** Sultan Almuntashiri.

**Formal analysis:** Sultan Almuntashiri, Yazed S. Alsowaida, Duo Zhang.

**Funding acquisition:** Sultan Almuntashiri, Yazed S. Alsowaida.

**Investigation:** Sultan Almuntashiri.

**Methodology:** Sultan Almuntashiri, Yazed S. Alsowaida, Duo Zhang, Ali Alghubayshi, Muteb Alanazi, Tareq Nafea Alharby, Mukhtar Ansari.

**Project administration:** Sultan Almuntashiri, Yazed S. Alsowaida.

**Resources:** Sultan Almuntashiri.

**Software:** Sultan Almuntashiri.

**Supervision:** Sultan Almuntashiri, Yazed S. Alsowaida.

**Validation:** Sultan Almuntashiri.

**Writing – original draft:** Sultan Almuntashiri, Yazed S. Alsowaida, Duo Zhang, Ali Alghubayshi, Muteb Alanazi, Tareq Nafea Alharby, Mukhtar Ansari.

**Writing – review & editing:** Sultan Almuntashiri, Yazed S. Alsowaida, Duo Zhang, Ali Alghubayshi, Muteb Alanazi, Tareq Nafea Alharby, Mukhtar Ansari.

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
