## [Decision Letter · Decision Letter 0]

3 Mar 2026

PONE-D-25-66274ICU-acquired infections drive higher mortality in community-acquired pneumonia–induced sepsis: insights from a transcriptomic analysisPLOS One

Dear Dr. Ansari

Thank you for submitting your manuscript to PLOS ONE. After careful consideration, we feel that it has merit but does not fully meet PLOS ONE’s publication criteria as it currently stands. Therefore, we invite you to submit a revised version of the manuscript that addresses the points raised during the review process.

**ACADEMIC EDITOR:**

Dear Authors,

Thank you for submitting your manuscript on ICU-acquired infections and 30-day mortality in pneumonia-induced sepsis, integrating transcriptomic analysis. This is an important and clinically meaningful topic. The attempt to connect clinical outcomes with gene expression patterns is a real strength of the study, and the comparison between community-acquired and hospital-acquired pneumonia adds valuable context.

After carefully considering the reviewer’s comments and assessing the manuscript myself, I agree that the study addresses a relevant question. At the same time, there are several methodological issues that need to be addressed before the conclusions can be considered robust.

The central concern relates to confounding. Mortality comparisons are largely based on unadjusted survival analyses. Given the retrospective design and the relatively small number of ICU-acquired infection cases, it is difficult to determine whether ICU-acquired infection independently predicts mortality or simply reflects greater baseline severity. If severity variables such as SOFA, APACHE II, age, or comorbidities are available, incorporating them into a multivariable model would substantially strengthen the analysis. If this is not feasible, the limitations should be clearly acknowledged and the conclusions reframed accordingly. In particular, any language implying causality should be moderated.

The small number of events, especially within subgroup analyses, also warrants caution. Borderline p-values and subgroup findings should be interpreted carefully. Providing effect sizes with confidence intervals and explicitly discussing the limited statistical power would improve transparency.

Regarding the transcriptomic component, additional clarification would be helpful. Please describe how gene expression data were normalized, whether batch effects were assessed, and how the genes of interest were selected. Because multiple genes were evaluated, the issue of multiple comparisons should be addressed, either through appropriate correction or through clear justification. If the findings are exploratory, this should be stated explicitly.

The definition of ICU-acquired infection also needs clearer description, including diagnostic criteria and timing relative to ICU admission. This information is essential for readers to interpret the findings appropriately.

Overall, the study is potentially interesting and hypothesis-generating. With stronger statistical adjustment, clearer methodological reporting, and more cautious interpretation, it could make a meaningful contribution. I encourage you to revise the manuscript carefully and provide a detailed response to the reviewer’s comments.

I look forward to reviewing your revised submission.

We look forward to receiving your revised manuscript.

Kind regards,

Gurmeet Singh, M.D., Ph.D.,

Academic Editor

PLOS One

Journal Requirements:

“This research is funded by Scientific Research Deanship at University of Ha’il- Saudi Arabia through project number BA-24 029.”

4. We note that your Data Availability Statement is currently as follows: [All relevant data are within the manuscript and its Supporting Information files]

6. Please upload a copy of Figure 1 & 2, to which you refer in your text. If the figure is no longer to be included as part of the submission please remove all reference to it within the text.

7. Please include a copy of Table 1 & 2 which you refer to in your text.

Additional Editor Comments (if provided):

Dear Authors,

Thank you for submitting your manuscript on ICU-acquired infections and 30-day mortality in pneumonia-induced sepsis, integrating transcriptomic analysis. This is an important and clinically meaningful topic. The attempt to connect clinical outcomes with gene expression patterns is a real strength of the study, and the comparison between community-acquired and hospital-acquired pneumonia adds valuable context.

After carefully considering the reviewer’s comments and assessing the manuscript myself, I agree that the study addresses a relevant question. At the same time, there are several methodological issues that need to be addressed before the conclusions can be considered robust.

The central concern relates to confounding. Mortality comparisons are largely based on unadjusted survival analyses. Given the retrospective design and the relatively small number of ICU-acquired infection cases, it is difficult to determine whether ICU-acquired infection independently predicts mortality or simply reflects greater baseline severity. If severity variables such as SOFA, APACHE II, age, or comorbidities are available, incorporating them into a multivariable model would substantially strengthen the analysis. If this is not feasible, the limitations should be clearly acknowledged and the conclusions reframed accordingly. In particular, any language implying causality should be moderated.

The small number of events, especially within subgroup analyses, also warrants caution. Borderline p-values and subgroup findings should be interpreted carefully. Providing effect sizes with confidence intervals and explicitly discussing the limited statistical power would improve transparency.

Regarding the transcriptomic component, additional clarification would be helpful. Please describe how gene expression data were normalized, whether batch effects were assessed, and how the genes of interest were selected. Because multiple genes were evaluated, the issue of multiple comparisons should be addressed, either through appropriate correction or through clear justification. If the findings are exploratory, this should be stated explicitly.

The definition of ICU-acquired infection also needs clearer description, including diagnostic criteria and timing relative to ICU admission. This information is essential for readers to interpret the findings appropriately.

Overall, the study is potentially interesting and hypothesis-generating. With stronger statistical adjustment, clearer methodological reporting, and more cautious interpretation, it could make a meaningful contribution. I encourage you to revise the manuscript carefully and provide a detailed response to the reviewer’s comments.

I look forward to reviewing your revised submission.

Reviewers' comments:

Reviewer's Responses to Questions

**Comments to the Author**

1. Is the manuscript technically sound, and do the data support the conclusions?

Reviewer #1: Partly

Reviewer #2: No

2. Has the statistical analysis been performed appropriately and rigorously? 

Reviewer #1: No

Reviewer #2: No

3. Have the authors made all data underlying the findings in their manuscript fully available?

Reviewer #1: Yes

Reviewer #2: Yes

4. Is the manuscript presented in an intelligible fashion and written in standard English?

Reviewer #1: Yes

Reviewer #2: Yes

5. Review Comments to the Author

Reviewer #1: This manuscript presents a secondary transcriptomic analysis of the GSE65682 dataset exploring the impact of ICU-acquired infections on 30-day mortality in pneumonia-induced sepsis, with subgroup analysis of CAP and HAP patients. The topic is clinically relevant, and the integration of outcome analysis with gene expression profiling is a strength of the study.

- The conclusions occasionally imply mechanistic or causal relationships (e.g., immune dysregulation “may predispose” to ICU-acquired infections). Given the retrospective design and single time-point gene expression measurement at ICU admission, the findings should be framed more explicitly as associative rather than predictive or causal.

- Survival comparisons were performed using Kaplan–Meier analysis without multivariable adjustment. With only 20 ICU-acquired infection cases (11 in the CAP subgroup), confounding remains a concern. Please clarify whether severity indices (e.g., SOFA, APACHE II, if available in GSE65682) were examined, and discuss how the lack of adjustment may influence interpretation.

- Five genes were tested without correction for multiple comparisons. Even with a limited gene set, statistical correction (e.g., Bonferroni or FDR) should be considered or justified. The IL7R finding (p=0.04 overall; p=0.01 in CAP) may not remain significant after strict correction.

- The operational definition of ICU-acquired infection within the dataset is not clearly described. Please specify timing criteria, diagnostic definitions, and how these events were identified in the secondary dataset.

- The CAP subgroup with ICU-acquired infection includes only 11 patients. The borderline mortality difference (p=0.05) should be interpreted cautiously. A brief discussion of limited statistical power and potential instability of estimates would strengthen the manuscript.

- Minor grammatical issues are present (e.g., “To evaluated the impact…” in the abstract).

- Please clarify whether gene expression data were normalized and whether batch effects were assessed or corrected.

- The discussion could be slightly condensed, particularly sections reiterating IL7R literature.

- Consider tempering claims regarding novelty of CAP vs HAP differential mortality unless supported by comprehensive literature evidence.

Reviewer #2: Thank you for the opportunity to review this manuscript examining the association between ICU acquired infections and 30 day mortality in pneumonia induced sepsis, together with related immune transcriptomic patterns. The topic is clinically important, and the integration of clinical outcomes with gene expression analysis is a meaningful and potentially valuable approach. The stratified analysis between community acquired pneumonia and hospital acquired pneumonia is particularly interesting and may have clinical implications. However, several methodological and interpretative concerns should be addressed before the manuscript can be considered suitable for publication in PLOS ONE.

The most important concern relates to confounding and the absence of multivariable analysis. Mortality comparisons rely primarily on Kaplan Meier curves and log rank testing, without adjustment for baseline severity, organ dysfunction, comorbidities, or other relevant ICU factors. Patients who develop ICU acquired infections are often more severely ill at baseline, and without proper adjustment it is not possible to determine whether ICU acquired infection independently predicts mortality or simply reflects greater illness severity. I strongly recommend performing multivariable Cox regression or logistic regression adjusting for available severity markers. Without such analysis, the conclusion that ICU acquired infections drive mortality should be softened to indicate association rather than causation.

The relatively small number of patients who developed ICU acquired infections also raises concerns regarding statistical power. Only twenty patients developed ICU acquired infections overall, and only eleven in the community acquired pneumonia subgroup. The reported p value of 0.05 in this subgroup is borderline and should be interpreted cautiously. The manuscript would benefit from reporting effect sizes with confidence intervals and explicitly acknowledging the limited precision of subgroup analyses.

Regarding the transcriptomic analysis, the rationale for selecting the five immune related genes requires clarification. Since the dataset is genome wide, it is important to specify whether these genes were prespecified based on biological hypotheses or selected after exploratory screening. In addition, no correction for multiple testing is described. Given the number of genes and subgroup comparisons performed, adjustment for multiple comparisons would strengthen the robustness of the findings. While IL7R appears statistically significant, the remaining genes did not reach significance and should be clearly described as exploratory trends.

The biological discussion of IL7R is thoughtful and supported by prior literature. The link to immune suppression in sepsis is plausible and clinically relevant. However, translational implications should be presented more cautiously given the retrospective and hypothesis generating nature of the analysis. Functional validation and independent cohort replication would be necessary before proposing biomarker guided immune targeted strategies.

The manuscript would also benefit from clearer reporting of how ICU acquired infection was defined within the dataset. Diagnostic criteria, timing of onset, and confirmation methods should be described to improve transparency and reproducibility.

The study has several strengths. The use of a publicly available dataset promotes transparency. The stratified analysis between community acquired pneumonia and hospital acquired pneumonia adds clinical nuance. The authors acknowledge several limitations. However, additional limitations should be emphasized, including lack of multivariable adjustment, small subgroup size, potential reverse causality, and absence of external validation.

Minor issues include a grammatical error in the abstract where “To evaluated” should read “To evaluate,” and inconsistency in referring to 28 day versus 30 day mortality. These should be corrected for clarity.

In summary, the manuscript addresses an important clinical question and presents potentially interesting findings, particularly regarding IL7R expression in community acquired pneumonia induced sepsis patients who develop ICU acquired infections. However, the current analysis does not fully support the strength of the conclusions. I recommend major revision, with particular attention to multivariable adjustment, clarification of gene selection strategy, correction for multiple testing, clearer definition of ICU acquired infection, and moderation of causal language.

6. PLOS authors have the option to publish the peer review history of their article (what does this mean?). If published, this will include your full peer review and any attached files.

Reviewer #1: No

Reviewer #2: No

---

## [Author Response · Author response to Decision Letter 1]

9 Apr 2026

We appreciate the thoughtful review remarks from the esteemed reviewers. The amended manuscript appears good, and we have addressed the criticisms made during the review process.

---

## [Decision Letter · Decision Letter 1]

22 Apr 2026

PONE-D-25-66274R1ICU-acquired infections drive higher mortality in community-acquired pneumonia–induced sepsis: insights from a transcriptomic analysisPLOS One

Dear Dr. Ansari,

Thank you for submitting your manuscript to PLOS ONE. After careful consideration, we feel that it has merit but does not fully meet PLOS ONE’s publication criteria as it currently stands. Therefore, we invite you to submit a revised version of the manuscript that addresses the points raised during the review process.

Thank you for submitting the revised version of your manuscript. The reviewers appreciate the improvements made, particularly in terms of clarity and overall presentation, and the study addresses a clinically relevant question with a meaningful integration of clinical and transcriptomic data. While both reviewers find the manuscript potentially suitable for publication, some concerns remain regarding the strength of interpretation relative to the data. To proceed toward acceptance, the interpretation of the findings should be further tempered to reflect the exploratory nature of the analysis, with avoidance of causal language, and the discussion should more explicitly acknowledge key limitations, including the absence of multivariable adjustment, the potential for confounding, the relatively small sample size, and the lack of correction for multiple testing. In addition, statements regarding IL7R should be reframed as hypothesis-generating, and the possibility of reverse causation should be more clearly addressed. As a recommendation, you may also consider further clarifying the exploratory scope of the transcriptomic analysis to improve overall transparency. Overall, the manuscript is progressing well and, with these revisions, should be well aligned with the strength of the data and suitable for publication.

We look forward to receiving your revised manuscript.

Kind regards,

Gurmeet Singh, M.D., Ph.D.,

Academic Editor

PLOS One

**Journal Requirements:**

**Additional Editor Comments:**

Thank you for submitting the revised version of your manuscript. The reviewers appreciate the improvements made, particularly in terms of clarity and overall presentation, and the study addresses a clinically relevant question with a meaningful integration of clinical and transcriptomic data. To further strengthen the manuscript prior to acceptance, the interpretation of the findings could be slightly tempered to better reflect the exploratory nature of the analysis, particularly by avoiding causal language and ensuring that conclusions remain proportionate to the strength of the data. It would also be helpful to more explicitly elaborate on key limitations, including the absence of multivariable adjustment and the potential influence of confounding factors such as disease severity, comorbidities, or ICU-related interventions, as well as the relatively small sample size and associated uncertainty in the estimates. In addition, a clearer emphasis on the exploratory nature of the transcriptomic analysis, including a brief note on the absence of correction for multiple testing, would improve transparency. The discussion of IL7R may also benefit from being framed as hypothesis-generating, and the possibility of reverse causation could be acknowledged more explicitly. Overall, the manuscript is progressing well and, with these refinements, should be well aligned with the strength of the data and suitable for publication.

Reviewers' comments:

Reviewer's Responses to Questions

**Comments to the Author**

1. If the authors have adequately addressed your comments raised in a previous round of review and you feel that this manuscript is now acceptable for publication, you may indicate that here to bypass the “Comments to the Author” section, enter your conflict of interest statement in the “Confidential to Editor” section, and submit your "Accept" recommendation.

Reviewer #1: (No Response)

Reviewer #2: All comments have been addressed

2. Is the manuscript technically sound, and do the data support the conclusions?

Reviewer #1: Partly

Reviewer #2: Partly

3. Has the statistical analysis been performed appropriately and rigorously? 

Reviewer #1: No

Reviewer #2: Yes

4. Have the authors made all data underlying the findings in their manuscript fully available?

Reviewer #1: Yes

Reviewer #2: Yes

5. Is the manuscript presented in an intelligible fashion and written in standard English?

Reviewer #1: Yes

Reviewer #2: Yes

6. Review Comments to the Author

**Reviewer #1:** This manuscript addresses an important and clinically relevant question regarding the impact of ICU-acquired infections on mortality in pneumonia-induced sepsis, while also exploring associated immune-related gene expression patterns using a publicly available dataset. The integration of clinical outcomes with transcriptomic data is a notable strength and adds value to the study. The revised version demonstrates improved clarity and a more appropriate acknowledgment of key limitations. The authors’ effort to refine the interpretation compared to earlier versions is appreciated. However, several important methodological constraints remain, and further refinement in the interpretation and framing of the findings is needed to ensure that the conclusions are proportionate to the strength of the data.

Major Comments:

1. Interpretation of mortality findings

The reported association between ICU-acquired infections and mortality, particularly in the CAP subgroup, is based on relatively small sample sizes and borderline statistical significance. In addition, the effect estimates are accompanied by wide confidence intervals, suggesting limited precision.

While the interpretation has been moderated in the revised version, the conclusions may still be perceived as stronger than supported by the data.

Suggestion:

The authors are encouraged to further emphasize that these findings represent unadjusted associations and to explicitly acknowledge the uncertainty related to limited statistical power and imprecision of the estimates. Any causal language should be avoided, and the interpretation should remain appropriately cautious.

In addition, the use of causal wording (e.g., “drive higher mortality”) in the title and conclusions should be reconsidered, as the study design supports association rather than causation.

2. Lack of multivariable adjustment

The authors appropriately note that multivariable analysis was not feasible due to dataset limitations. However, this remains a critical limitation, as important confounding factors such as illness severity, ICU interventions, and comorbidities are not accounted for.

Suggestion:

This limitation should be more prominently emphasized in the Discussion. The authors should clearly state that the observed associations may be influenced by unmeasured confounding and should therefore be interpreted with caution.

3. Transcriptomic analysis and multiple testing

The manuscript appropriately describes the gene expression analysis as exploratory. However, the absence of correction for multiple comparisons should be more explicitly justified, as this increases the risk of false-positive findings.

Suggestion:

The authors should reinforce that these findings are hypothesis-generating and avoid implying confirmatory or clinically actionable conclusions. A brief justification for not applying multiple testing correction would improve transparency.

4. Interpretation of IL7R as a potential biomarker

The discussion suggests that IL7R may serve as a potential biomarker. Given the exploratory nature of the analysis and the absence of external validation or predictive performance assessment, this interpretation may be premature.

Suggestion:

The authors are encouraged to reframe IL7R as a candidate signal requiring further validation, rather than a clinically applicable biomarker at this stage.

5. Reverse causality

Gene expression was measured at ICU admission, while ICU-acquired infections occurred later during the ICU stay. Therefore, the temporal relationship between gene expression and infection risk remains uncertain.

Suggestion:

The Discussion would benefit from a more explicit consideration of potential reverse causality and a clear statement that causal inference cannot be established from the current analysis.

Minor Comments:

1. Please ensure consistency in reporting mortality outcomes (28-day vs 30-day), as inconsistencies are currently present in the Abstract and Results sections.

2. The rationale for selecting the candidate genes should be clarified (e.g., whether they were pre-specified based on prior evidence).

3. Consider reporting effect sizes (e.g., fold changes) alongside p-values for gene expression results to improve interpretability.

4. Minor language editing is recommended to improve clarity and readability.

Overall, the manuscript has merit and addresses a relevant clinical question. With further refinement in interpretation and clearer positioning of the findings as exploratory and hypothesis-generating, it would be suitable for publication after minor revisions.

**Reviewer #2:** Thank you for the revised manuscript. The authors have made several meaningful improvements, particularly in clarifying the methodology, providing effect size estimates, and moderating the interpretation of the findings. These revisions have strengthened the manuscript and addressed several of the concerns raised in the previous round.

However, a few important issues remain. The absence of any adjusted analysis continues to be a key limitation. Even without formal severity indices, adjustment for available baseline variables such as age, sex, and diabetes would still be feasible and would help clarify whether ICU-acquired infection is an independent predictor of mortality or simply a marker of more severe illness. In addition, the small sample size, particularly in the subgroup of CAP patients with ICU-acquired infection, substantially limits statistical power, and the borderline findings should be interpreted with caution.

While the transcriptomic analysis is now described as exploratory, the potential impact of multiple testing and the associated risk of false-positive findings could be more explicitly discussed. The rationale for selecting the specific candidate genes would also benefit from further clarification, particularly regarding whether this approach was hypothesis-driven.

Overall, the manuscript has improved and addresses a clinically relevant question, but further clarification and more cautious interpretation of the findings would strengthen the validity of the conclusions.

7. PLOS authors have the option to publish the peer review history of their article (what does this mean?). If published, this will include your full peer review and any attached files.

Reviewer #1: No

Reviewer #2: **Yes:** Steffi Nata, MD

---

## [Author Response · Author response to Decision Letter 2]

29 Apr 2026

We have addressed all of the insightful comments raised during second round of the review, attached the 'response to reviewers' file in the submission system

---

## [Editor Report · Decision Letter 2]

6 May 2026

ICU-acquired infections and mortality in community-acquired pneumonia–induced sepsis: insights from a transcriptomic analysis

PONE-D-25-66274R2

Dear Dr. Ansari

We’re pleased to inform you that your manuscript has been judged scientifically suitable for publication and will be formally accepted for publication once it meets all outstanding technical requirements.

Kind regards,

Gurmeet Singh, M.D., Ph.D.,

Academic Editor

PLOS One

Additional Editor Comments (optional):

Dear Authors,

Thank you for submitting the revised version of your manuscript.

The reviewers’ comments have been addressed satisfactorily, and the manuscript has improved considerably compared to the previous version. In particular, the interpretation of the findings is now more balanced, the exploratory nature of the transcriptomic analysis has been clarified appropriately, and the study limitations are better acknowledged throughout the manuscript.

Although several limitations remain, particularly related to the retrospective design, small subgroup sample size, and limited adjustment for confounders, these issues have been discussed adequately and do not preclude publication.

The study addresses a clinically relevant topic and provides useful exploratory data regarding ICU-acquired infections and immune-related gene expression patterns in pneumonia-induced sepsis.

I am pleased to inform you that your manuscript is accepted for publication in PLOS ONE.

Sincerely,

Academic Editor

PLOS ONE
---

## [Editor Report · Acceptance letter]

PONE-D-25-66274R2

PLOS One

Dear Dr. Ansari,

I'm pleased to inform you that your manuscript has been deemed suitable for publication in PLOS One. Congratulations! Your manuscript is now being handed over to our production team.

Kind regards,

on behalf of

Dr. Gurmeet Singh

Academic Editor

PLOS One